# Infiltration by Intratumor and Stromal CD8 and CD68 in Cervical Cancer

**DOI:** 10.3390/medicina59040728

**Published:** 2023-04-07

**Authors:** Polina Dimitrova, Mariela Vasileva-Slaveva, Velizar Shivarov, Ihsan Hasan, Angel Yordanov

**Affiliations:** 1Department of Pathology, Medical University-Pleven, 5800 Pleven, Bulgaria; 2Department of Breast Surgery, Shterev Hospital, 1000 Sofia, Bulgaria; 3Research Institute, Medica University Pleven, 5800 Pleven, Bulgaria; 4Department of Obstetrics and Gynecology, University Hospital “Sofiamed”, 1750 Sofia, Bulgaria; 5Department of Gynecologic Oncology, Medical University-Pleven, 5800 Pleven, Bulgaria

**Keywords:** cervical cancer, CD8, CD68, intratumoral, stromal

## Abstract

*Background and Objectives:* The tumor microenvironment (TME) plays a major role in neoplastic development. Various types of cells can be found in the TME. These cells can be classified into two groups, immunosuppressive and immunostimulatory types, depending on the function they perform in the antitumor immune response (IR). By interacting both with each other and with tumor cells, different immune mechanisms are activated or inhibited, which can suppress or promote the development and progression of cervical cancer (CC). Our aim was to investigate some of the main components of the cellular immune response in TME—tumor-infiltrating cytotoxic T cells (Tc, CD8+) and tumor-associated macrophages (TAMs, CD68+)—in patients with CC. *Materials and Methods:* We analyzed 72 paraffin-embedded tumor tissues of patients diagnosed and treated at Medical University Pleven, Bulgaria. Patients were classified according to the 2018 FIGO (International Federation of Gynaecology and Obstetrics) classification. From each patient, we selected one histological slide with hematoxylin eosin staining. In a microscopic evaluation, CD8+ T lymphocytes and CD68+-positive macrophages were counted in the tumor and stroma of five randomly selected fields at ×40 magnification (HPF). We analyzed the relationship between intratumoral and stromal CD8 and CD68 expression and FIGO stage and N status. *Results:* There was no significant association between the expression levels of intratumoral and stromal CD68+ cells in the different FIGO stages and according to the lymph nodes’ involvement. For CD8+ cells, the association of stromal infiltration was also not found, but T intratumor infiltration was associated with a higher FIGO stage, despite the fact that the results did not reach significance (*p* = 0.063, Fisher test). Intratumoral CD8+ cells were significantly associated with positive N status, (*p* = 0.035). *Discussion:* The separation of tumor-infiltrating cytotoxic T cells and tumor-associated macrophages into intratumoral and stromal is inconsequential. In our study, the level of infiltration of CD68+ cells in tumors and stromata was not significantly associated with tumor progression or lymph node involvement. The results were different for CD8+ cells, in which levels of infiltration were associated with lymph nodes’ statuses. *Conclusions*: The separate evaluation of CD68+ immune cells in the TME as intratumoral and stromal is not beneficial for defining prognoses, since the presence of these cells is not associated with the patient’s stage. In our study, the presence of CD8+ cells was significantly associated with lymph node metastases. The prognostic value of the obtained results can be enriched with an additional study of the lymphocyte phenotype, including B and other subtypes of T lymphocytes, NK cells, as well as molecules involved in the immune response, such as HLA subtypes.

## 1. Introduction

Despite the available options for the prevention and treatment of patients with cervical cancer (CC), including screening for human papilloma virus (HPV), prophylactic HPV vaccines, the surgical resection of already existing tumors, radiotherapy and chemotherapy, the incidence of the disease remains high worldwide [1]. The conventional methods used for its treatment lead to a wide range of side effects, relapses and short disease-free survival [2,3].

The tumor microenvironment (TME) plays a major role in neoplastic development. Various types of cells are found in it, including tumor-associated fibroblasts, natural killer (NK) cells, macrophages, dendritic cells and T and B lymphocytes. These cells can be classified into two groups, immunosuppressive and immunostimulatory types, depending on the function they perform in the antitumor immune response (IR) [4]. By interacting both with each other and with tumor cells, different immune mechanisms are activated or inhibited, which can suppress or promote the development and progression of CC.

Immunotherapy has led to a revolution in the treatment of various malignant diseases. The final goal of its application is to direct the body’s immune system against tumor cells. In October 2021, the U.S. Food and Drug Administration approved pembrolizumab in combination with chemotherapy or bevacizumab as a first-line treatment for CC [5]. Investigating the amount and spatial localization of key immune cell participants such as CD8+ and CD68+ cells will contribute to a better understanding of antitumor IR and its prognostic value in this type of cancer. The obtained results could enrich the information needed to improve the effect of the approved treatment or to develop another specific immunotherapy [2].

The natural immune response relies on the interaction of adaptive and innate immunity systems and the synergy between them. Among the tumor immune cell infiltrates, cytotoxic T cells (Tc, CD8+) in the adaptive immune system have been found to act specifically on cancer cells and are the most powerful effectors in the anticancer immune response. Tumor-associated macrophages (TAMs, CD68+) are elevated in many cancers. They are able to influence the activity of other immune cells, creating a pro or antitumor microenvironment. In studies with carcinomas of different localizations, CD8 and CD68 levels have been found to have conflicting prognostic value [6,7,8,9,10,11]. Since the data on this topic are few in patients with CC, we set out to study the role of some of the main components of the cellular immune response in TME—Tc, CD8+ and TAMs and CD68+ in this neoplasm—using an immunohistochemical method.

The aim of this research was to study some of the main components of the cellular immune response in TME—Tc and TAMs—in patients with CC in different stages and to analyze the relationship between intratumoral and stromal CD8 and CD68 expression and FIGO stage and N status.

## 2. Materials and Methods

### 2.1. Patients

This is a retrospective study which included patients diagnosed with CC in the university hospitals of Medical University Pleven, Bulgaria for a 6-year period of time between 1 January 2015 and 31 December 2020. The cases studied were randomly selected (until the predetermined number was dialed—72) from archival lists of the Department of Pathology of these hospitals. We used materials available in the hospital pathology archive that contained a sufficient amount of tumor tissue and in which the study would not threaten their depletion or impairment.

The 2018 FIGO (International Federation of Gynaecology and Obstetrics) classification was used [12] (Table 1). There were only patients with lymph node involvement in FIGO stage III, because patients with FIGO stage IIIA and FIGO stage IIIB are not candidates for primary surgery. In all patients, radical hysterectomy with lymph node dissection was performed without previous treatment.

The patients’ characteristics were obtained from the medical records and are given in Table 1.

### 2.2. Immunohistochemical Examination

For each patient participating in the study, one histological slide with hematoxylin eosin staining was selected. Slides for the immunohistochemical (IHC) assessment of the subtypes of immune cells—Tc and macrophages—were prepared from its corresponding formalin-fixed paraffin-embedded (FFPE) tissue specimen.

The EnVision™ FLEX, High pH, (Link), DAKO visualization system was used for antibody detection. From FFPE tissue samples, tissue sections with a thickness of 2–4 µm were prepared and placed on adhesive slides. Each of the 72 studied patients was tested with each of the two primary antibodies (pre-diluted, ready to use) and a visualization system using an AutostainerLink 48 automated system, DAKO (Table 2). All work procedures (including the deparaffinization and rehydration of tissue sections; heat-induced antigen retrieval; incubation with primary antibody and with the detection system; the visualization of the complex, etc.) when conducting the IHC analysis were performed according to the protocols for the respective antibodies of the manufacturing company. In each staining run, external control tissues were used to establish the functionality of the staining reagents, to assess the quality of the staining reaction, to determine the expression pattern of the antibodies used and to optimize the IHC work procedures before applying them to the studied cases (Table 3, Figure 1 and Figure 2).

The scoring of positive IHC-stained cells (colored brownish—with membrane and/or cytoplasmic expression for Tc; cytoplasmic for TAM), was performed independently by two pathologists who were not aware of the clinicopathological data of the studied cases. Similar to another study of ours [13], immunophenotyped lymphocytes (separately intratumoral and stromal) were counted (computer-assisted) and semiquantitatively graded. The average value of the results of the investigated cases with CC was taken as the cut-off.

In microscopic evaluation, CD8+ T lymphocytes and CD68+-positive macrophages were counted in the tumors and stromata of five randomly selected fields at ×40 magnification (HPF). Results with a mean number of up to 25 IHC-positive cells in stromal localization and up to 12 or 13 in intratumoral localization (for CD8 and CD68 subtypes, respectively) were considered as a low value of the respective cell subtype. With the corresponding higher number of positive cells, their high intratumoral or stromal concentration was reported (Figure 3 and Figure 4).

### 2.3. Statistical Analysis

Because of the small sample size, distributions of cases within each subgroup were analyzed using two-sided Fisher’s exact test. Uncorrected *p*-values below 0.05 were considered significant. Analyses were performed using base statistical functions from R v. 4.2.2 for Windows. Plots were generated using ggpubr v. 0.6.0 package for R.

## 3. Results

The expression levels of intratumoral and stromal CD8 and CD68 in patients with different stages of CC are shown on Table 4. There were very few or no samples in some of the groups, especially in the group with missing CD8+ and CD68 cells. Generally, the low concentration of the infiltration immune cells was most frequently observed in the advanced FIGO stage (FIGO III).

We analyzed the relationship between the levels of intratumoral and stromal CD8+ and CD68+ cells, FIGO stage and N status.

We compared the expression and the levels of intratumoral and stromal CD8+ cells in the different FIGO stages. Despite the fact that the results did not reach significance, there was a trend for a positive association between the CD8+ cells in the tumors and FIGO stage (*p* = 0.063, *p* = 1). When comparing the levels according to the different N status, we found a statistically significant difference in the infiltration of the tumors with CD8+ cells (*p* = 0.035). The infiltrations with CD8+ cells in the stroma were neither associated with FIGO stage nor with the N status (*p* = 0.870) (Figure 5).

The infiltrations of tumor and stroma with CD68+ cells were not associated with FIGO stage (*p* = 0.245, *p* = 0.162) and with N status (*p* = 0.185, *p* = 0.319) (Figure 6).

## 4. Discussion

Neoplasms are complex structures in which tumor cells interact with their surroundings during their progression (invasive growth and metastatic spread). The antitumor cytotoxic cellular response is performed with the main participation of antigen-presenting cells and CD8+ cytotoxic T lymphocytes [2].

Macrophages are part of innate immunity, and through their ability to process and present antigens to produce cytokines necessary for T-lymphocyte activation, they are critical for initiating and mediating specific IR as well [14]. They play an important role in the fight against infections and neoplasms, but also in the regulation of the metabolic response to tissue stress [15]. There is a growing body of research investigating the role of macrophages in carcinogenesis, the prognostic and predictive role of their presence in the inflammatory infiltrate of TME and the possibility of macrophage-targeted immunotherapy [16,17].

The most common risk factor associated with the occurrence of CC is human papillomavirus (HPV) infection [2]. One of the hallmarks of oncogenic-virus-associated inflammation has been found to be the attraction and activation of a monocyte-macrophage cell population to the TME [17]. There are few reports on the role of CD68+ macrophages in CC, and the data on their prognostic significance for prognosis and survival are controversial [17,18,19].

According to some studies, the number of macrophages increases with the progression of cervical neoplastic lesions [16,18,20,21,22,23,24]. Tumor cells have been suggested to secrete various types of cytokines into the TME to promote monocyte migration to the neoplastic tissue. Moreover, a high concentration of these cells correlates with unfavorable clinical findings, such as metastatic lymph nodes and a more advanced stage according to FIGO. A possible explanation is that under the influence of tumor cells, macrophages probably contribute to carcinoma progression by changing their phenotype to one with reduced antigen-presenting function and the suppression of T-cell proliferation [16,17,25]. Other studies have found that high numbers of TAMs show a significant negative correlation with tumor stage and are not a prognostic marker in CC [1,17,26]. Immune cell quantification alone does not reflect the dynamics and functionality of the TME. In viable tissues, they are not fixed and can move spatially.

Similar to studies in other neoplasms, immune cells can be divided into intratumoral (intraepithelial) and stromal. Cells in the tumor nest or in direct contact with neoplastically altered cells are defined as intratumoral. Stromal immune cells in the connective tissue component pass between the tumor (parenchymal) zone without having direct contact with the tumor cells. Both types are true tumor-infiltrating immune cells—i.e., they are located in the tumor area. Their distinction is conditional, because it reflects their static localization in the studied histological section [27]. According to some results, the stromal component demonstrates a higher number of macrophages than the intraepithelial one. However, increased numbers of macrophages in the epithelial zone have also been found to be associated with malignant transformation in cervical cancer [17].

However, macrophages represent a phenotypically heterogeneous group of cells, and their physiology can be significantly modified in response to various biochemical factors in the TME [16]. Macrophages can be divided into two main categories—M1 and M2. M1 or classical macrophages are responsible for distinguishing and destroying various pathogens and cancer cells. They apply two mechanisms for effective outcomes: (i) directly mediating cytotoxicity to kill cancer cells, e.g., by using tumor-killing reactive nitrogen and oxygen species (ROS and NO) molecules [28]; (ii) antibody-dependent cell-mediated cytotoxicity (ADCC) [2,29]. Intraepithelial infiltration with high numbers of M1 macrophages is an independent prognostic factor for a favorable prognosis [17].

M2 macrophages have a weakened capacity for antigen presentation and therefore have low antitumor activity. They have an increased capacity for tissue remodeling. They are associated with the progression and regulation of angiogenesis in CC [30]. M2 macrophages promote epithelial–mesenchymal transition (EMT), which is a hallmark of invasion and metastasis, thereby promoting CC progression. Macrophage polarization to the M2 phenotype correlates with a reduced response to chemo- and radiotherapy and short survival in patients with regionally advanced cervical carcinoma. It has been found that after contact with tumor cells, M1 macrophages can undergo transformation from an M1 to an M2 phenotype [2,31], which is mainly associated with progressive tumors [16,25,26].

One of the shortcomings of our study is the use of a common CD68 macrophage marker, which cannot distinguish M1 or M2 subtypes of infiltrating macrophages. This does not allow us to give more definite and specific results about the prognostic value of this type of immune cells [17].

For a better understanding of the antitumor IR in CC, it is necessary to study its other main participants—tumor-infiltrating immune lymphocytes (TILs) [2]. Their presence in tumors is a reflection of the dynamic interaction of the host’s immune system with tumor antigens and the tumor microenvironment (TME). TILs represent a heterogeneous population of immune cells in tumor tissues. Functionally different B lymphocytes, NK and T cell subtypes may be present.

A growing number of studies have shown that the degree and composition of the tumor IR are prognostic for many solid malignant neoplasms. TILs’ assessment is recommended as a biomarker for routine histopathology reports in some cancers, but not yet in CC [1]. The analysis of the type, location, number and ratio of TILs (indicating phenotypic changes) may also provide useful information on the progression of CC and prognosis related to response to treatment.

CD8+ cytotoxic T cells are major contributors to the destruction of cancer cells [2]. CD8+ TILs can kill tumor cells directly and indirectly. In the direct mechanism, they recognize specific tumor antigens and secrete factors such as perforin and granzyme, causing tumor cell death. In the indirect mechanism, they induce the apoptosis of the neoplastic cell population [1,32,33]. CD4+ T cells are the ones that provide help to CD8+ cells. The two cell types suggest primary immunity to tumor cells and are considered together in most studies of TILs in CC.

The established results show that the concentration of different types of immune cells could be used to predict the treatment outcome in patients with CC [3]. Conflicting data are available regarding the prognostic value of immune cells in different tumor regions [3,4,5,6,7,8,9,10,11,12,13,14,15,16,17,18,19,20,21,22,23,24,25,26,27,28,29,30,31,32,33,34,35,36,37,38,39,40,41,42,43,44,45,46,47].

A trend toward increased TILs levels with the increasing grade of neoplastic epithelial changes has been established [1,34,35]. A detailed analysis of the literature on T-lymphocyte subpopulation studies found that in the epithelial layer of an HPV+ normal cervix, in severe dysplasia and in cancer progression, the concentrations of CD8+ TILs were significantly higher than CD4+. In the stroma, CD4+ TILs predominated in all groups of progression of epithelial changes, but no statistical difference was found between them. Because HPV mainly infects epithelial cells, these data suggest that CD8+ TILs are on the front line fighting virus-infected and neoplastically altered cells [1,3]. In early-stage cancer, CD8+ cells and the CD8+/CD4+ ratio were also found to be significantly elevated in patients without lymph node metastases. When the CD8+/CD4+ ratio decreases, both overall and disease-free survival rates decrease [44,45,46,47]. Some authors found an increase in the number of both types of T cells—CD8 and CD4—with the progression of the degree of the lesion [34]. According to data from other studies, there is a directly proportional correlation between a high number of CD4 and CD8 lymphocytes [36,37] and its regression [5,38]. The largest number of studies have shown decreased expression levels of both T-lymphocyte subpopulations in precancerous lesions and CC, emphasizing the importance of local immunosuppression on the evolution of HPV-induced changes [16].

The evaluation of TILs in tumors is also becoming increasingly important in the search for the ideal biomarker to select patients with the highest probability of response to applied therapy, including immunotherapy. Some studies have evaluated the relationship of TILs with the results of chemotherapy or radiotherapy in patients with CC. According to them, an abundance of CD8+ TILs was associated with a better response to treatment. The presence of CD8+ TILs has been suggested to be a potential independent favorable prognostic factor for patients with cervical adenocarcinoma after radiotherapy. Patients with CD8+ TIL in tumor foci had a significantly better OS compared with patients without CD8+ TIL infiltration [1,3].

In our study, we compared the expression of stromal and intratumoral CD8+ and CD68+ cells, and we analyzed their role in tumor progression. We only found a significant association of the infiltration of CD8+ cell in tumors with the presence of metastatic lymph nodes.

Based on this, we cannot prove the role of intratumoral or stromal CD8 and CD68 expression in the progression of CC.

Unfortunately, the sample size of our study was relatively small, which probably influenced the small number of significant dependencies found. In addition, the carcinomas included in the study were not divided into groups depending on their histological type.

## 5. Conclusions

The separate evaluation of CD68+ immune cells in the TME as intratumoral and stromal is not beneficial for defining prognoses, since the presence of these cells is not associated with the patient’s stage. In our study, only the presence of CD8+ cells was significantly associated with lymph node metastases. The prognostic value of the obtained results can be enriched with an additional study of the lymphocyte phenotype, including B and other subtypes of T lymphocytes, NK cells, as well as molecules involved in the immune response, such as HLA subtypes.

## Figures and Tables

**Figure 1 medicina-59-00728-f001:**
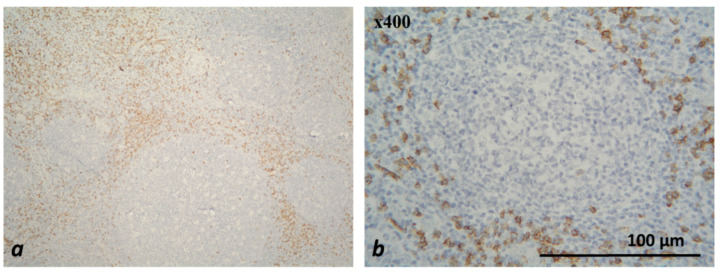
Microscopic evaluation of IHC-stained CD8+ T-lymphocyte subtypes (colored brownish) in tonsil—×200 (**a**), ×400 (**b**).

**Figure 2 medicina-59-00728-f002:**
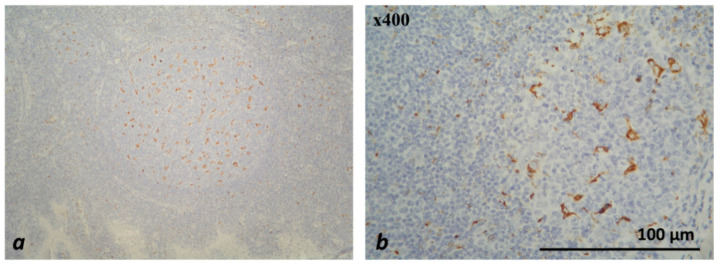
Microscopic evaluation of IHC-stained CD68+ macrophages (colored brownish) in tonsil—×200 (**a**), ×400 (**b**).

**Figure 3 medicina-59-00728-f003:**
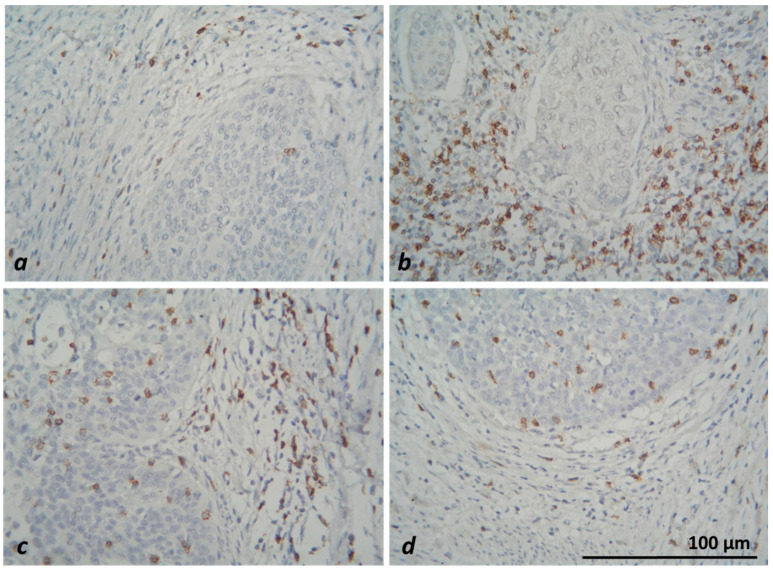
Microscopic evaluation of IHC-stained CD8+ T-lymphocyte subtypes (colored brownish) in CC—low intratumoral and stromal (**a**), low intratumoral and high stromal (**b**), high intratumoral and stromal (**c**) and high intratumoral and low stromal (**d**) concentration of positive cells, ×400.

**Figure 4 medicina-59-00728-f004:**
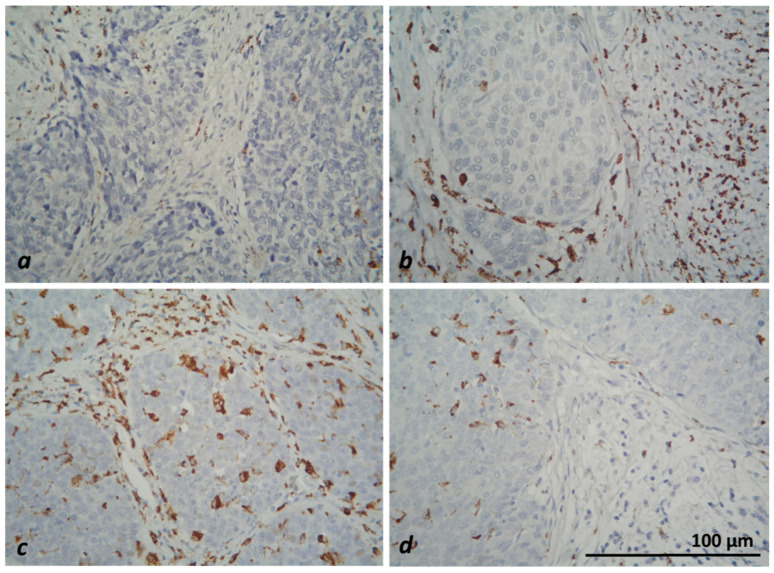
Microscopic evaluation of IHC-stained CD68+ macrophages (colored brownish) in CC—low intratumoral and stromal (**a**), low intratumoral and high stromal (**b**), high intratumoral and stromal (**c**) and high intratumoral and low stromal (**d**) concentration of positive cells, ×400.

**Figure 5 medicina-59-00728-f005:**
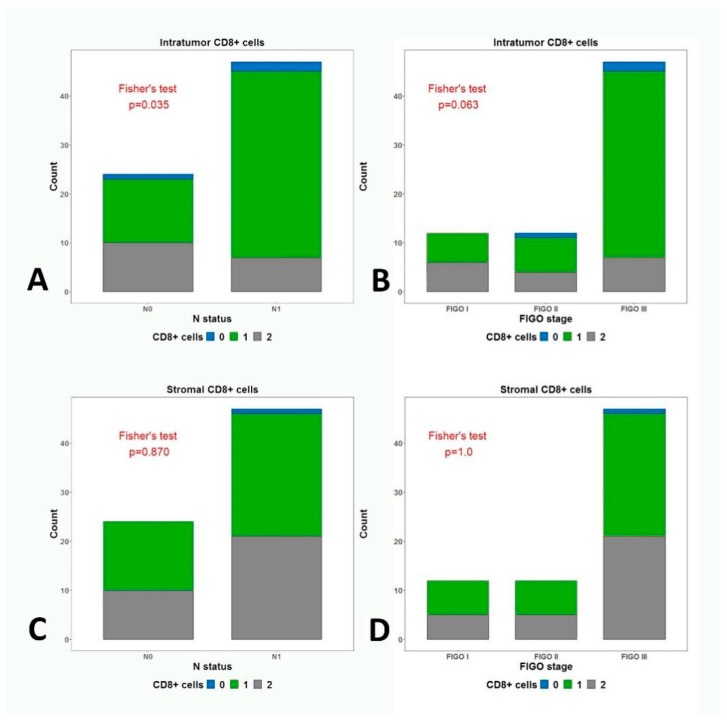
Fisher’s test for the association between (**A**) intratumor CD8 expression and N status; (**B**) intratumor CD8 expression and FIGO stage; (**C**) Stromal CD8 expression and N status; (**D**) stromal CD8 expression and FIGO stage.

**Figure 6 medicina-59-00728-f006:**
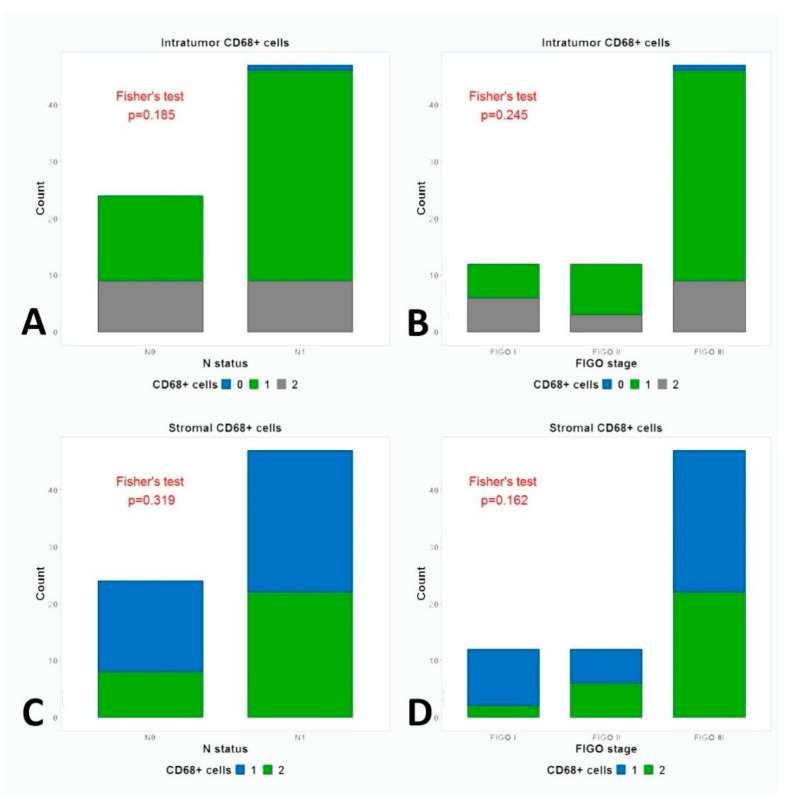
Fisher’s test for the association between (**A**) intratumor CD68 expression and N status; (**B**) intratumor CD68 expression and FIGO stage; (**C**) Stromal CD68 expression and N status; (**D**) stromal CD68 expression and FIGO stage.

**Table 1 medicina-59-00728-t001:** Characteristics of patients.

Patients Characteristics	*N*	%
FIGO 1	13	18.06
FIGO 2	12	16.66
FIGO 3	47	65.28
N0	25	34.72
N1	47	65.28
Total	72	100

**Table 2 medicina-59-00728-t002:** Supplies used.

Primary Antibody	Clone	Company	Solutions	Visualization System	Platform	Cellular Specificity
CD8, Mo	C8/144B, Isotype: IgG1, kappa	Dako, Agilent, Glostrup, Denmark	Ready to use	EnVision™ FLEX, High pH, (Link), DAKO	AutostainerLink 48, DAKO	Cytotoxic T cells
CD68, Mo	PG-M1, Isotype: IgG3, kappa	Dako, Agilent, Glostrup, Denmark	Ready to use	EnVision™ FLEX, High pH, (Link), DAKO	AutostainerLink 48, DAKO	Macrophages

**Table 3 medicina-59-00728-t003:** External control tissues used and IHC expression pattern established for respective antibodies.

Primary Antibody	Control Tissue	Reaction Location in the Cells	Positive Quality Control
CD8	Tonsil	Membrane and/or cytoplasm	Moderate to strong intensity of expression in T cells of the interfollicular areas; expression is absent in B cells and germinal centers
CD68	Tonsil	Cytoplasm	Moderate to strong intensity of expression in germinal center’s macrophages

**Table 4 medicina-59-00728-t004:** Distribution of patients according to FIGO stage and immunohistochemical findings.

Stage Histologic Results	FIGO Stage I	FIGO Stage II	FIGO Stage III
Intratumoral CD8			
missing	0	1	2
low concentration	6	7	38
high concentration	7	4	7
Stromal CD8			
missing	0	0	1
low concentration	7	7	25
high concentration	6	5	21
Intratumoral CD68			
missing	0	0	1
low concentration	7	9	37
high concentration	6	3	9
Stromal CD68			
missing	0	0	0
low concentration	11	6	25
high concentration	2	6	22

## Data Availability

The authors declare that all related data are available from the corresponding author upon reasonable request.

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
