# Peer review of "Infiltration by Intratumor and Stromal CD8 and CD68 in Cervical Cancer"

_medicina, 2023, doi:10.3390/medicina59040728_

Round 1
Reviewer 1 Report
Dimitrova et al. reported the expression of intratumor and stromal CD8 and CD68 in cervical cancer. The study could provide more insights into the tumor micro environment study, and useful for cancer treatment. However, I have the following concerns for the current manuscript.
The results part is too simple. Design and purpose of the study should be stated. It is not clear what the authors wanted to show us. Analysis of results should be also provided, rather than simply listing the data.
The conclusions are not clear. The authors only compared the expression levels of CD8 and CD68, without further experiments to show the significance of these findings.
Most of the discussion part is not related to the results shown in the manuscript.
Reviewer 2 Report
Overall, the idea of the work is interesting. The work results can have merit in the related field. However, several essential concerns should be addressed.
In general, the manuscript needs English polishing.
Abstract
The Abstract is well structured and shortly presents the results and impact of the study. Just the acronym “CC” for cervical cancer should be expanded at its first mention in the abstract.
Introduction
This section needs to be more expanded regarding the association of the selected tumor infiltrating cytotoxic T-cells (Tc, CD8+) and tumor-associated macrophages (TAMs, CD68+) with the cellular immune response in TME of other cancers. The authors should give a strong evidence-based rational for selecting these types of cells among others to be investigated in the present work.
Materials and methods
This section needs extensive revision:
- Are the patients included in the current analysis "consecutive"? Please add a Consort Flow diagram to graphically describe the inclusion criteria and reason for exclusion of some pts.
- “FIGO” acronym should be expanded, and a supportive reference should be cited.
- The duration of sample collection should be clear in this section.
- The place from which the participants were selected should be detailed.
- Did any participant receive any type of treatment before taking part?
- What about the demographic and basic characteristics (e.g. age, FH, risk factors,…etc.) of the included participants?
- Did the authors take the characteristics of the included participants by themselves or they revised the medical records?
- From where did the authors obtain the FFPE samples?
- Did they obtain the ethical approval and patient consent before taking part?
- The applied concentration (dilution factor) of all antibodies used should be provided.
Fig. 1 and 2:
- How did the authors control the intra-tumor heterogeneity?
- To be more informative, all figures should be supported by pointing to the required data from the figure and provide the key for the different cells and color in the figure legend as not all future readers are provided to be specialty-related ones.
- Also, a scaling bar should be provided in all figures.
- Where is the statical analysis section?
Results
- As sample number in some table cells less than 5, the authors should justify the study sample size and the study power in the statistical analysis section.
- Figures 5 and 6 color code is not clear. Also, a space should be left between the upper vs. the lower panels to avoid reader confusion.
- In general, a note should be added if p values were corrected for multiple testing.
Discussion
- page 8; the 3rd line: “and the data on their prognostic significance are controversial (11).” This should be supported by more than one reference and the controversy should be expanded and clarified to the readers.
- Also, in the next section, “According to some studies…” should be supported by more than one reference.
- “FIGO (International Federation of Gynaecology and Obstetrics), should be mentioned earlier at its first mention in the manuscript, not here!
- Why the font size in this section is not consistent throughout the text?
- The authors should provide the study limitation(s) by the end of the discussion.
Author Response
please see the atachment

Round 2
Reviewer 1 Report
Thanks for the follow up. The authors have addressed my concerns in the previous review, and I don't have further suggestions on the current manuscript. I would support its publication in the present form.
Reviewer 2 Report
The authors have adequately addressed the concerns raised by the reviewer. Thank you